# Conjugated Polymeric Materials in Biological Imaging and Cancer Therapy

**DOI:** 10.3390/molecules28135091

**Published:** 2023-06-29

**Authors:** Qinbin Zheng, Zhuli Duan, Ying Zhang, Xinqi Huang, Xuefan Xiong, Ang Zhang, Kaiwen Chang, Qiong Li

**Affiliations:** 1Shandong Provincial Key Laboratory of Detection Technology for Tumor Markers, College of Medicine, Linyi University, Linyi 276005, China; 2College of Chemistry and Chemical Engineering, Linyi University, Linyi 276005, China; 3Key Laboratory of Medical Molecular Probes, Department of Medical Chemistry, School of Basic Medical Sciences, Xinxiang Medical University, Xinxiang 453003, China

**Keywords:** conjugated polymer, cell imaging, tumor diagnosis and treatment

## Abstract

Conjugated polymers (CPs) have attracted much attention in the fields of chemistry, medicine, life science, and material science. Researchers have carried out a series of innovative researches and have made significant research progress regarding the unique photochemical and photophysical properties of CPs, expanding the application range of polymers. CPs are polymers formed by the conjugation of multiple repeating light-emitting units. Through precise control of their structure, functional molecules with different properties can be obtained. Fluorescence probes with different absorption and emission wavelengths can be obtained by changing the main chain structure. By modifying the side chain structure with water-soluble groups or selective recognition molecules, electrostatic interaction or specific binding with specific targets can be achieved; subsequently, the purpose of selective recognition can be achieved. This article reviews the research work of CPs in cell imaging, tumor diagnosis, and treatment in recent years, summarizes the latest progress in the application of CPs in imaging, tumor diagnosis, and treatment, and discusses the future development direction of CPs in cell imaging, tumor diagnosis, and treatment.

## 1. Introduction

In recent years, with the progress of society and the rapid development of industry, the incidence rate of tumor diseases has continued to rise, posing a serious threat to people’s health and becoming a major killer [1,2]. How to deal with such diseases has become a core issue in the key fields of biomedicine [3,4]. The detection of disease-related biomolecules and the early diagnosis of diseases are of great significance, and the quantitative detection of these markers with high sensitivity, high selectivity, and simple portability is particularly important [5,6,7]. In the progress of tumor diagnosis and treatment, traditional diagnosis and treatment methods, such as computed tomography (CT), magnetic resonance imaging (MRI), chemotherapy, and radiotherapy, have their own shortcomings, including high side effects, low visual resolution, and low sensitivity, as well as complex operation and drug resistance to tumors, which are greatly limited, so the development of a new type of sensitive, efficient, and non-invasive early diagnosis and treatment of cancer is urgent [8,9,10]. Understanding the pathogenesis of malignant tumors provides a strong impetus for the rapid development of intelligent tumor diagnosis and treatment strategies. With the development of new materials and breakthroughs in the performance of various advanced materials, tumor imaging, diagnosis, and treatment have developed rapidly. At present, various microspheres, microcapsules, liposomes, and polymer nanoparticles have been proven to be effective tools for drug delivery, biological imaging, and treatment [11,12,13,14]. In addition, material-based nanoassembly has paved the way for addressing many uncertain challenges in cancer imaging and treatment, including biocompatibility, biochemical diversity, high loading efficiency of hydrophilic and hydrophobic therapeutic agents, and the ability to passively and actively target cancers [15,16,17].

CPs are a new type of fluorescent material with excellent optical properties and multifunctional chemical structures. By regulating the structure of CPs, CPs with different photophysical properties can be prepared, providing a new development field for biosensing and biomedical research [18,19,20,21]. CPs are composed of a large number of conjugated repeat units and have unique photoelectric properties, which are characterized by high brightness, extinction coefficient, light stability, energy conversion efficiency, and low cytotoxicity [22,23,24,25]. From the perspective of structure, CPs have a hydrophobic backbone with a delocalized π electron structure, which mainly determines their light absorption and emission ability, signal transmission ability, and fluorescence quantum yield. By regulating the structure of the backbone, polymers with different absorption and emission wavelengths can be prepared. The regulation of the backbone can also be achieved by doping some conjugated monomers or changing the proportion and arrangement of the repeating units on the backbone. According to the different backbones of the polymer chain, it can be divided into the following types: polyfluorene (PF), polythiophene (PT), poly(fluorene-phenylene) (PFP), polypyrrole (PPy), polyparaphenylene (PPP), poly(phenylene-ethynylene) (PPE), poly(p-phenylenevinylidene) (PPV), and polydiacetylene (PDA). By modifying the side chains with different functional groups, they can be used to regulate the interactions between polymers and biological macromolecules, cells, or bacteria. In addition, functional groups, such as drug molecules, photosensitizers, and specific recognition units, can also be connected at the end of the side chain. Through continuous in-depth research, CPs with special functions have been synthesized, providing new materials and broadening their applications in the field of biomedicine [26,27,28,29,30,31,32,33,34].

Over the past decade, we have witnessed the enormous potential of CP materials in the fields of chemistry, medicine, life sciences, and materials science, mainly due to their unique photochemical and photophysical properties [35,36]. CPs have a unique electronic structure, and each light-emitting unit has a molecular line effect through electronic delocalization and electronic coupling. Specifically, when a conjugated polymer with strong light capture ability is excited, the generated excitons can quickly transfer along the long-range electron skeleton to the energy acceptor, converting small fluctuations in the environment into changes in the conjugated polymer fluorescence signal, thereby achieving highly sensitive detection of the detected object [37,38,39]. In addition, CP materials utilize their signal amplification effect and energy transfer mechanism (FRET), which can be combined with biological recognition elements to achieve high selectivity and sensitivity in the detection biomolecules [40,41].

With the expansion of conjugated polymer functions, conjugated polymer nanoparticles and conjugated polymer dots have developed well in imaging, sensing, and treatment technologies [42,43]. Because of their good biocompatibility, large absorption coefficient, excellent light stability, rich functional groups, low biological toxicity, and continuously adjustable optical and electrical properties, they can also be used as a photoacoustic imaging (PAI) contrast agent, gaining them more research interest [44]. Conjugated polymer nanoparticles and conjugated polymer dots usually show broad absorption spectra, and the absorption peaks are located in the near-infrared (NIR) window. Due to their diversity of structure and function, they can also be used as a visual imaging probe for PAI. It is worth noticing that, compared with other types of PAI probes, conjugated polymer nanoparticles usually show higher light stability and enhanced PA brightness [45]. In addition, conjugated polymer nanoparticles can also be used as photothermal reagents, due to their high NIR light absorption and strong non-radiation quantum efficiency [46]. In the NIR window, the photothermal effect of conjugated polymer nanoparticles has the ability to convert light energy into heat, which can be used to ablate tumor cells for photothermal therapy (PTT) [47,48]. Conjugated polymer dots are easy to couple with biological molecules, can participate in the energy transfer process with other dyes, and can be used in complex biological environments. They also have outstanding security, stability, and resolution, which provide better materials for cell imaging. As excellent biomedical nanoparticles, conjugated polymer nanoparticles and conjugated polymer dots have been used to develop a range of biomedical diagnostic and therapeutic methods.

In this review, we summarize recent advances in the design and application of CP materials, mainly in bioimaging and tumor diagnosis and treatment. We focus on problems of bioimaging and cancer therapy of CPs, with an emphasis on the discussion of researchers’ strategies to rationally design functionalized CPs for bioimaging and cancer diagnosis and treatment. Firstly, we summarized the applications to obtain biological information, utilizing functionalized CPNPs with high brightness and good photostability, such as multicolor imaging, two-photon imaging, super-resolution imaging, photoacoustic imaging, and Raman imaging, and discussed potential challenges and developments for the future. Additionally, based on the properties of tunable emission wavelengths and photothermal and photoacoustic conversion, various cancer therapies have been developed. We illustrated the potential mechanisms of photoconversion and reviewed the progress of emerging phototherapies, including photothermal and photodynamic therapy and combination therapy. Finally, the potential challenges and future developments of CP materials for tumor diagnosis and treatment are presented to improve the optical functionalization of CP materials, as well as their translation to clinical applications. CPs are expected to be developed as a multifunctional, intelligent, and minimally invasive material to treat a wide range of malignancies, with excellent therapeutic outcomes.

## 2. Results

### 2.1. Bioimaging

As a powerful visualization tool, imaging technology has been widely discussed because of its potential application in the biomedical field. In the past, significant progress has been made with imaging probes and nanoparticles [49,50,51]. Due to their excellent sensitivity, biocompatibility, and outstanding optical properties, CPs have been developed in many areas of biological imaging [52,53,54].

#### 2.1.1. Multicolor Imaging

Multicolor imaging is commonly used in research to better describe disease features and understand biological processes. For example, in immuno-oncology, multicolor imaging allows for the study of different immune cells relative to key biomarker locations. In neuroscience, multicolor imaging enables the simultaneous study of multiple proteins. Quantum dots (QDs) are, so far, the most successful, due to their tunable optical properties in the spatial dimension, but their composition of heavy metals has limited their clinical application [18]. Conjugated polymer nanoparticles (CPNPs), which combine the characteristics of CPs and nanoparticles, are a promising new class of fluorescent material that is being integrated into the interdisciplinary fields of materials chemistry, biology, and medicine. Since CPNPs avoid the problem of heavy metal toxicity and display significant optical properties in biological imaging, they show great potential in this field [55,56,57,58]. The structural design of CPs allows for absorption spectra and fluorescence emission covering the entire visible and near-infrared (NIR) regions, thus further facilitating the development of multicolor imaging. However, CPNPs are limited by the available range of colors, which means that the application of multispectral imaging is limited. Feng et al. synthesized CPs with multiple fluorescence emissions using fluorene and thiophene derivatives (Figure 1a) [59]. By nanoassembly, the synthesized CPs with multiple fluorescence emission were assembled with poly(styrene maleic anhydride) (PSMA) to obtain multicolor CPNPs with high quantum yield that exhibited multicolor emission under excitation at the same wavelength. After binding to antibodies, these immunolabeled CPNPs could be used for the specific imaging of cancer cells through multispectral imaging. Due to the limitation of the available color range of particles, different CPs are required for multispectral imaging. Therefore, polychromatic CPNPs were obtained through the self-assembly of multiple polymers (Figure 1b). However, different polymers have different chemical structures, which can affect their optical and physical properties when they are wrapped. Green et al. tuned the visible spectrum emission of a single polymer type (poly[2-methoxy-5(2′-ethylhexyloxy)-1,4-phenylenevinylene], MEH-PPV) by using an oxidation method, keeping the synthesized conjugated polymer brightly fluorescent and displaying a color change from blue to red, depending on the amount of oxidant used, which was applied to cell imaging [60]. Compared to other monochromatic fluorescent nanocarriers, multicolor CPNPs showed better specificity for cancer cells. Highly delocalized fluorescent conjugated polymers (FCPs) can be fine-tuned for absorption and emission spectra by modification of the main chain or side chain, providing potential for multicolor imaging. Due to the ease with which hydrophobic FCPs can be dissolved in nano micelles and improve fluorescence brightness through concentration and size effects, nanoparticles containing FCPs may also have potential in biological imaging. Wang et al. successfully synthesized a novel amphiphilic cellulose-graft-poly(p-diox-anone) (MCC-graft-PPDO) and combined FCPs with different emissions into MCC-graft-PPDO for tumor cell imaging [61]. When incubated with liver cancer cells and breast cancer cells, bright fluorescence was observed in the cytoplasmic region due to effective uptake of FCPs/MCC-graft-PDDO, indicating that multicolor micelles containing FCPs could be a potential tool for tumor cell imaging and diagnosis. Therefore, the synthesis of nanomaterials with multicolor fluorescence has achieved better targeted imaging and specific detection.

#### 2.1.2. Near-Infrared Imaging

In the past few years, great progress has been made in the visible fluorescent imaging of cells [62,63]. Inspired by the D-A structure design, researchers have improved the penetration depth and sensitivity of biological imaging in biological tissues and living animals by changing the structure of CPs’ main chain or side chain to extend the fluorescence emission spectrum to the NIR region [64,65,66]. Liu et al. first reported FR/NIR fluorescent CPs (PFBTDBT) composed of fluorene and benzothiadiazole through effective intermolecular and intramolecular energy transfer, providing an attractive fluorophore for in vivo imaging at an emission wavelength of 698 nm with a PL quantum yield of 0.27. They have been shown, through in vivo studies, to be effective FR/NIR fluorescent probes for targeted in vivo fluorescence imaging and tumor detection [67], having good cell compatibility and providing basic guidance for the further design and development of new FR/NIR probes for potential clinical applications. In addition, with low biological autofluorescence and high tissue penetration depth in the FR/NIR region compared to the visible region, they usually exhibit lower fluorescence quantum yields due to strong stacking and intramolecular charge transfer in aqueous solution. To overcome this limitation, new FR/NIR-emitting CPs with greatly improved fluorescence quantum yields were designed and used for highly specific targeted cancer imaging [68]. Ultra-small nanoparticles (CPNPs) were successfully prepared with FR/NIR fluorescent conjugated polymers as the core and amphiphilic phospholipid polyethylene glycol (DSPE-PEG) as the shell protection layer. The ultra-small CPNPs, with a diameter of 6 nm, have very good stability in water solutions. CPNPs with a larger size of 30 nm were also synthesized and used as a control group. The fluorescence quantum efficiency of ultra-small nanoparticles is as high as 41%, which is much higher than the 25% of large nanoparticles, greatly improving the clarity and resolution of imaging. In another study, due to the limited depth of light penetration and the variation in brightness of the fluorescent probe marker, Wu et al. designed highly fluorescent polymer dots with FR light absorption and near-infrared (NIR) emission for cell tracking [69]. The highly fluorescent polymer dots were designed to produce narrow-band emission upon FR uptake, with a quantum yield of approximately 22%, and were coated with cell-penetrating peptides, resulting in a significant increase in cell labeling brightness, enabling in vivo tracking and revealing unique cell distribution and migration actions [70]. Zhang et al. synthesized NIR fluorescent conjugated polymer dots using AIE fluorophores as cross-conjugated side chains to induce fluorescence emission [71]. Dual-color cell imaging can be performed at 540 nm visible light and 716 nm NIR light with a quantum yield of 31.8%, which can be applied to in vivo cell toxicity analysis and microvascular imaging. This dual-color function provides a method for in vivo cell tracking or tissue imaging, and provides potential application prospects for various biological imaging, monitoring, and drug delivery applications. Subsequently, fluorescence emission further expanded from NIR-I to NIR-II (1000–1700 nm). Dai et al. used single-walled carbon nanotube NIR-II photoluminescence for noninvasive skull-brain imaging with high spatial resolution below 10 μm and depth-resolved brain vasculature exceeding 2 mm in fluorescence imaging mode [72]. Huang et al. synthesized the conjugated polymer TTQ-2TC (triazole [4,5-g]-quinoline (TTQ)) with good absorption performance in the NIR-II region by introducing long alkyl side chains decorated with dithiophene (2TC), which can dissolve in solution. This polymer exhibits good water dispersibility and has a bright NIR-II FI signal (Figure 2a) [73]. Subsequently, photosensitive nanoparticles (TTQ-2TC NPs) were prepared by a simple nanoprecipitation method. TTQ-2TC NPs can achieve high-resolution angiography of the hind limbs and abdominal vessels. Under 1064 nm laser irradiation, they can generate a high-temperature effect and have significant anti-tumor effects, suggesting the potentiation of NIR-II FI imaging and NIR-II PTT nanodrugs in tumor treatment (Figure 2b). The reduced photon scattering in this spectral region enhances the penetration depth and signal-to-noise ratio of fluorescence imaging.

#### 2.1.3. Two-Photon Imaging (TPFI)

Due to the two-photon excitation in NIR light, two-photon microscopy has become a powerful tool in fluorescence imaging, providing many key advantages over conventional fluorescence techniques using single-photon excitation, such as low spontaneous fluorescence and high penetration depth [74]. However, the development of two-photon absorption (TPA) probes is still limited by severe cytotoxicity, a small TPA cross-section, and a low quantum yield. Therefore, it is highly necessary to develop new fluorescent probes to overcome these drawbacks. McNeill et al. reported that CPs have multiphoton properties and, further, used CP nanoparticles for multiphoton fluorescence imaging, which showed great potential in the field of multiphoton microscopy techniques [75]. Subsequently, Xu et al. used a two-photon-excited fluorescence resonance energy transfer mechanism to demonstrate that cationic CPs can significantly enhance the two-photon-induced emission of DNA embedding agents [76]. In addition to the utilization of polymer and biomolecule assembly strategies, Schanze et al. reported the development of an effective method to enhance green fluorescent protein (GFP) expression by using cationic polystyrene (PPE) as a two-photon absorption sensitizer [77]. Wang et al. reported a polymer (PTA5) composed of dibenzothiophene-S,S-dioxide derivatives. As an effective photosensitizer, PTA5 has a relatively bright green emission and a satisfactory quantum yield [78]. PTA5 NPs are prepared by coating a biocompatible polymer matrix. Upon excitation at 800 nm, PTA5 in aqueous solutions showed a broad two-photon absorption cross-section, good tissue penetration depth, and optical stability, making them suitable for biological imaging (liver vessels and mouse ear vessels). To expand the TPA cross-section, CPs with dual photoactivity were integrated into nanoparticles for real-time visualization of vascular imaging. Tan et al. reported a novel CP with a poly(fluorenone-co-thiophene) skeleton, which exhibits excellent biocompatibility, photostability, lysosomal co-localization, and two-photon imaging properties, and has deeper tissue penetration compared to single-photon confocal imaging [79]. Subsequently, Tang et al. reported a type of nanoparticle (AIE NPs) that can aggregate and induce[80]. Upon excitation at 800 nm, they can generate high quantum yields and maximum cross-sectional absorption, as well as singlet oxygen (^1^O_2_) and •OH. In a mouse melanoma model, it has been observed that AIE NPs can accumulate in tumor blood vessels, demonstrating that AIE NPs can be used as imaging agents for spatiotemporal imaging of tumor tissue (Figure 3). With further exploration by researchers, two-photon imaging and deep visualization of the vascular system showed that CPs have potential prospects for TPFI. Liu et al. developed near-infrared region II (NIR-II) photoexcited single-chain CPNPs with bright fluorescence in the near-infrared region I (NIR-I) window for in vivo deep two-photon fluorescence imaging of intact mouse brains, and showed a high signal-to-noise ratio at 1200 nm excitation [81]. In addition, TPFI was performed on intact skulls, and reconstructions of brain vasculature were obtained at larger vertical depths, further confirming the great potential of bright NIR fluorophores for deep-tissue imaging in vivo. To improve their photostability, Liu et al. proposed a new core-shell method, using SiO_2_ as a coating to protect CPNPs and further improve their photostability [82]. PFBT-SiO_2_ NPs were obtained by reacting poly(9,9-dihexylfluorene-alt-2,1,3-benzothiadiazole) (PFBT) with SiO_2_. The synthesized PFBT NPs have good colloidal stability, photostability, and biocompatibility, and were successfully applied to two-photon excited cerebral angiography and further used for high-contrast cerebral vascular visualization in mice, confirming that the PFBT-SiO_2_ NPs prepared by the co-protection strategy have an excellent TPA cross-section and high fluorescence quantum yield (QY). For mouse brain and cranial TPFI, PFBT-SiO_2_ overcomes the limitation of poor penetration depth in optical microscopy and further facilitates studies in mouse brain imaging, as well as cranial bone imaging (Figure 4). In addition to the ability to perform deep-tissue imaging in the NIR emission, NIR light with good penetration capability can be used as an excitation source. In recent years, TPFI has been the most widely used NIR-excited optical imaging technique, which is a powerful tool for studying the structure and function of organs in living animals, especially for brain research in neuroscience.

#### 2.1.4. Super-Resolution Imaging

Over a long period of time, optical microscopy has gradually developed into one of the most important imaging tools in the life science field [83]. However, the spatial resolution of conventional fluorescence microscopy is limited by classical optical diffraction, which greatly hinders the resolution of subcellular structures and the observation of dynamic processes [84,85]. With the revolutionary progress in super-resolution fluorescence microscopy, the diffraction limitation has been overcome, providing unprecedented visualization of the subcellular world and greatly accelerating critical biological innovations. To date, various super-resolution techniques have attracted great interest among researchers and have experienced tremendous development in the past few years. Therefore, these technologies bring great hope for the biological revolution and play a crucial role in improving the imaging quality of fluorescent probes used for super-resolution imaging. It has been reported that CPNPs with particle sizes larger than 15 nm, without significant fluctuations prepared by nanoprecipitation, can effectively overcome light scattering, and displayed stable fluorescence signals. However, fluorescence blinking was detected in small-sized CPNPs with a particle size of about 10 nm. Wu et al. achieved super-resolution imaging with small-sized light scintillation quantum dots, labeling subcellular structures using the fluorescence intermittency phenomenon of nanoparticles, and demonstrated their superiority as super-resolution imaging probes [86]. Small-sized blinking quantum dot fluorescent probes composed of PFBT and [2-methoxy-5-(2-ethylhexoxy)-1,4-(1-cyanvinylidene-1,4-phenylene)] (CN-PPV) were prepared, which showed yellow and red fluorescence, respectively, when excited at their corresponding excitation wavelengths, and showed good dispersibility in buffer solution. Under super-resolution optical fluctuation imaging (SOFI) microscopy, CPNPs exhibit significant fluorescence blinking effects, high fluorescence brightness, and good photostability. Therefore, immunolabeled blinking CPNPs have been successfully used for super-resolution visualization of various subcellular organelles. Although significant progress has been made with monochrome SOFI imaging using blinking CPNPs, multicolor super-resolution fluorescence imaging remains a crucial challenge for studying protein-protein interactions and multiple subcellular structures, as small-sized blinking CPNPs as traditional nanomaterials display very broad emission and severe spectral crosstalk. To address this, small-sized light scintillation CPNPs with different spectral windows and narrow fluorescence emission bands were used to achieve multicolor SOFI, demonstrating that these CPNPs have excellent labeling functions in monochromatic and bichromatic SOFI of specific subcellular structures [87]. They synthesized two small light scintillation CPNPs, PFO and PFTBT5 CPNPs, with relatively narrow blue and carmine emission windows, respectively. These CPNPs have high fluorescence brightness, good photostability, superior biocompatibility, and pronounced fluorescence fluctuations. The CPNPs were labeled on multiple organelles simultaneously using the biocoupling technique, successfully achieving super-resolution imaging of multiple organelles in a high-resolution manner. In addition, effective light scintillation modulation of CPNPs was achieved by adjusting the D-A ratio in the polymer, ultimately providing a spatial resolution of 95 nm at the subcellular level, an almost four-fold increase in resolution compared to conventional optical diffraction images. In contrast, expansion microscopy (ExM) offers a unique approach to visualizing nano-scale cellular structures, but challenges, such as fluorescence group degradation and dilution during sample amplification, limit its widespread application. Wu et al. designed multifunctional Pdots for ExM applications, utilizing the Pdots’ tunable optical properties and excellent optical characteristics to achieve cell targeting, gel anchoring, and high signal-to-noise ratio (SNR) fluorescence reporting in fluorescence imaging [88]. Pdots achieved various subcellular structures and neuronal synapse visualization at a spatial resolution of 80 nm, with outstanding brightness and effective labeling. Additionally, by combining ExM and SOFI, subcellular structures of about 30 nm were resolved on a conventional microscope, highlighting the ability of Pdots to promote ExM-enhanced super-resolution imaging in biological imaging (Figure 5).

#### 2.1.5. Photoacoustic Imaging (PAI)

PAI is widely used in the field of bioimaging because it can overcome penetration depth limitations by combining optical excitation with ultrasound detection. Compared with optical imaging techniques, PAI achieves imaging beyond the limit of optical diffusion, providing deeper tissue imaging penetration and higher spatial resolution. PA materials suffer from their own poor photostability and low biocompatibility defects [89,90]. Therefore, the development of PA materials with good photostability and biocompatibility is an urgent need to fully expand PAI applications.

In 2014, Rao et al. reported a new type of CPNPs after in-depth analysis and demonstrated that the NPs exhibited better photostability and a higher photoacoustic output signal in PA molecular imaging than previously reported materials [91]. They were used for lymph node PA imaging in live mice and were used to study ROS production processes through in vivo PA imaging. It has been reported that the ability of CPs to generate PA is closely related to their optothermal properties. Inspired by this, Pu et al. introduced ultra-small carbon dots (fullerenes) into CPNPs, inducing photo-induced electron transfer through mutual interactions under light irradiation [92]. With an increase in fullerene content, the fluorescence intensity of CPNPs decreased significantly, accompanied by an increase in PA intensity, thus highlighting the enormous potential of CPNPs in enhancing PA signals through molecular orbital engineering. The slow elimination rate of synthesized PA imaging agents after entering the body, low reaction signal specificity, and acute or chronic toxicity to major organs affected detection capabilities and hindered the development of clinical application research. Therefore, Pu et al. reported a fluorescent probe (FPRR) with high renal clearance and the ability to image disease biomarkers in kidney injury in real time [93]. The FPRR has a near-infrared fluorescence (NIRF) signal and a PA signal, enabling rapid, efficient, and sensitive detection of drug-induced acute kidney injury (AKI) biomarkers. After 24 h of drug injection, the drug was found to be excreted through the kidney, and the deep-tissue penetration ability of PAI resulted in a 2.3-fold higher ratio of probe signal to background than NIRF imaging. Activatable PAI probes were achieved for real-time, sensitive imaging of renal function at the molecular level. Since PAI is a hybrid imaging modality based on NIR excitation and thermal ultrasound detection, it has a greater ability to penetrate deeper tissues compared to fluorescence imaging (FLI). Notably, activatable PAI probes trigger signals in the presence of specific molecular biomarkers, providing measurable information on physiological and pathological states at the molecular level. Thus, activatable PAI probes have been extensively investigated for in vivo imaging of a variety of bioactive metabolites and disease-specific biomarkers. Pu et al. reported that an imaging probe (SPCy) alters the NIRF and PA signals of neutrophil elastase (NE) in real time [94]. SPCy is constructed by the self-assembly of amphiphilic semiconductor polymers (SPs) and half-quenched peptide-coupled phthalocyanine dyes with NE. Since the oxygen atoms in phthalocyanines have weak electron-donating abilities, their NIRF and PA signals are generally in the “off” state. When NE is present, the protective peptide bond is cleaved, followed by a spontaneous 1,6-elimination reaction that activates the NIRF and PA signals of phthalocyanines. However, the NIRF and PA signals of SPs remain unchanged. Therefore, SPCy can achieve ratio detection of NIRF and activatable PA imaging for tumor-associated neutrophils (TANs) in live mice (Figure 6). Cheng et al. reported CPNPs as a prostate-specific membrane antigen targeting probe for NIR-II prostate cancer imaging [95]. By introducing transient absorption (TA) microscopy and PAI, they studied, in detail, the distribution of nanoparticles in tumor tissues after intravenous injection of CPNPs. TA microscopy extended imaging probes from single cells to entire organs, providing sub-micrometer resolution and high sensitivity, further demonstrating the targeting of CPNPs to prostate-specific membrane antigen-positive prostate cancer. Subsequently, PA tomography showed good imaging depth and contrast, revealing the potential of CPNPs for deep-tissue imaging in vivo. Intravenous administration showed that prostate-specific membrane antigens can selectively accumulate and be retained in tumors, enabling non-invasive PA detection of prostate tumors in vivo. To fully decipher 3D biological structures with a large signal-to-noise ratio and deep penetration, optical resolution PA microscopy imaging (ORPAMI) has become an advanced imaging technique for 3D biological structures with a high signal-to-noise ratio and deep penetration. Therefore, new NIR-II CPNPs were first used as excellent contrast agents for assisting NIR-II ORPAMI. NIR-II CPNP-assisted ORPAMI can accurately locate the vascular system and tumors in the brain. CPNPs have ultra-high PA sensitivity, and the wide field of view allows clear observation of the regular vascular system in mouse ears. Due to the low drug delivery efficiency of existing cancer treatment methods and the lack of non-invasive, routine, and cost-effective postoperative monitoring methods, a comprehensive anticancer strategy combining the PA effect and polymer nanoparticles has been proposed. Inspired by this, Wu et al. used CPNPs as transport carriers, photothermal materials, and imaging contrast agents for delivering nanoparticles into solid tumors [96]. Due to the strong near-infrared absorption coefficient of nanoparticles, which provides rich external contrast for PAI, the distribution of nanoparticles can be visualized and monitored. The combination of an NIR pulse laser with accumulated nanoparticles provides significant therapeutic efficiency for photothermal therapy (PTT). In addition, endogenous hemoglobin enables PAI to assess the efficacy of tumor therapy through structural and functional parameters of the vascular network, and to monitor post-operative tumor progression in vivo, providing new opportunities for precise tumor treatment planning and timely feedback on treatment outcomes (Figure 7). In summary, photoacoustic imaging (PAI) is a promising imaging method that can visualize the brain vessels, skull, and whole-body tissues of mice, providing great potential for early diagnosis and treatment of tumors. At the same time, the excellent properties of PAI have also attracted numerous researchers to work hard on clinical transformation applications.

#### 2.1.6. Raman Imaging

Recently, the rapid development of optical microscopy has promoted research on biomedical applications. Raman microscopy, as a non-labeling imaging technology, has become a new tool for biological imaging. Due to the interference of its inherent small cross-section and endogenous background signals, Raman scattering cannot obtain high-contrast Raman images of cells and tissues [97,98,99]. Therefore, there is an urgent need to develop new probe materials with strong Raman signals in the Raman silence region (1800–2800 cm^−1^). However, current probes have very weak Raman vibrations, which are not suitable for immediate signal collection. To overcome these problems, Wang et al. pioneered the exploration of the excellent potential of CP materials to enhance Raman signals by using CPs with alkyne units as Raman active materials for cell imaging [100]. By screening the Raman signals of three cationic CPs, it was found that PPE exhibited a strong and narrow C-C triple bond characteristic vibration peak near the Raman silence region of 2200 cm^−1^. It was confirmed that the presence of alkyne in the conjugated backbone of PPE can effectively enhance the vibration of alkyne through the aromatic and delocalized conjugated structure. In addition, by using peptides as Raman probes to modify water-soluble cationic PPE through synergistic enhancement effects, Raman imaging of live cells in the Raman silent region was performed. This strategy not only demonstrates the good potential of enhancing Raman signals with CPs materials, but also expands the application of CPs in biological imaging. Tang et al. reported PPE materials with typical AIE characteristics, showing strong aggregated-state fluorescence, interesting self-assembly behavior, inherent enhancement of alkyne vibrations in the Raman silent region of cells, and effective antibacterial activity [101]. It not only solves the low fluorescence efficiency of traditional PPE conjugated polymers in the aggregated state, but also fully utilizes the inherent alkyne Raman signal enhancement effect of PPE polymers to realize dual enhancement effects of fluorescence and Raman signals, and serves as a fluorescent/Raman dual-mode imaging probe, sensitively displaying tumor areas and residual tumors, which is beneficial to accurate tumor resection surgery. Raman imaging has shown great potential as a powerful vibro-optical spectroscopic technique for the diagnosis of cancer and the visualization of various biological processes, but there are few reports on the use of polymers as Raman probes for in vivo imaging, mainly due to the weak Raman signal intensity of the polymers themselves and the strong fluorescent background generated under excitation that can affect the Raman signal. Some studies have reported that polymers doped with polydopamine burst the fluorescence and increase the Raman signal intensity of the polymer through an intermolecular energy transfer process. However, doping methods provide only limited Raman signal enhancement for polymers, and these doped polymer probes still do not allow for high resolution and sensitive in vivo imaging. Xiao et al. reported that poly(indine-dithiophene-benzothiadiazole) (IDT-BT) polymers as Raman probes emit strong Raman signals under excitation at 785 nm, achieving in vivo Raman imaging [102]. Due to the unique Raman properties of IDT-BT polymers, they avoid fluorescence interference by relying on molecular modulation and absorption edges adjacent to the desired excitation wavelength and emit strong Raman scattering upon resonant excitation of the characteristic signal molecules. Impressively, the IDT-BT Raman probes enable non-invasive microvascular imaging, which is not possible with other Raman probes. With the development of stimulated Raman scattering (SRS) microscopy, normally weak Raman scattering can be amplified by non-linear optical processes, allowing rapid label-free imaging of living cells and tissues. Wu et al. reported the introduction of a strong biologically orthogonal Raman-labeled alkynyl group into photochromic diarylethylene (DTE) and designed a photo-switched vibration probe [103]. When photoisomerization occurs under ultraviolet UV irradiation, the closed-loop reaction changes the electronic structure of the DTE unit, leading to a red shift in its absorption spectrum, which not only produces a large Raman shift of the conjugated alkynyl group, but also enhances its stimulated Raman scattering intensity through electronic pre-resonance effects (Figure 8). As a result, it can be reversibly switched on or off in controlled, visible light. In summary, Raman imaging technology has a low background signal, high spatial resolution, excellent optical stability, and non-invasive detection capabilities. Therefore, in tumor diagnosis, Raman imaging is a promising research tool that not only provides molecular-level information to distinguish between cancerous and non-cancerous cells and tissues, but also enables non-invasive microvascular imaging, allowing for rapid unlabeled imaging of living cells and tissues.

#### 2.1.7. Future Directions for Bioimaging

CPs is a new type of in vivo imaging probe developed in recent years that is of great significance for the early diagnosis of tumors (Table 1). The rapid development of a variety of in vivo imaging techniques based on CPs has greatly promoted the development of cancer diagnosis. However, the metabolic status of CPs and long-term toxicity studies in vivo need to be further elucidated, which is an important direction for future development, and is expected to provide strong evidence for clinical applications. Meanwhile, it is necessary to continue developing new conjugated polymers that can adapt to more complex biochemical environments, while retaining the functional group activity and photochemical and photophysical properties of the main chain structure, to play a role in deeper biological systems. Although CPs have been studied at the cellular and animal levels, most current biological imaging is still limited to the passive imaging or passive targeting stages of tumors. Therefore, the future direction of active targeted tumor imaging needs further exploration. In addition, it is also necessary to strengthen the development of multifunctional CPs and multimodal imaging. At present, research on multimodal imaging is limited to optical imaging, including fluorescence imaging and PA imaging. If we can strengthen the construction of multimodal CPs materials that cover optical imaging, MRI, CT, and PET imaging and use multifunctional CPs materials to assist in tumor diagnosis, it will be the next favorable development direction.

### 2.2. Tumor Diagnosis and Treatment

In recent years, although great achievements have been made in disease treatment, cancer remains a major challenge [115,116]. Understanding the pathogenesis and processes of malignancies will promote the development of new diagnostic and therapeutic strategies [117,118,119]. Currently, in clinical application, the main methods of cancer treatment are surgery, chemotherapy, and radiotherapy [120]. With the development of various disciplines, researchers have established a series of new anti-cancer methods. The cancer treatment strategy based on conjugated polymers provides a highly promising platform for tumor suppression and clearance, mainly including photodynamic therapy (PDT), photothermal therapy (PTT), and several synergistic therapy strategies (Table 2) [121,122,123,124,125].

#### 2.2.1. Photodynamic Therapy (PDT)

PDT is a progressive therapy that utilizes photodynamic effects to diagnose and treat diseases [141,142,143,144]. PDT is typically composed of three important parts: a specific wavelength of light, photosensitizer (ps), and oxygen [145]. Tissues can be excited by irradiation with light of a specific wavelength, which will excite the ps to transmit energy to surrounding oxygen, generating highly active ^1^O_2_ to react with adjacent biological systems and producing cell toxicity effects [146,147,148]. Therefore, the resulting cell toxicity induces cancer ablation through various mechanisms, including necrosis, apoptosis, and tumor microvascular damage. Generally, effective photosensitizers (PSs), appropriate light, and sufficient oxygen molecules (O_2_) are the basic factors for effectively producing ^1^O_2_. However, due to critical issues, such as light penetration depth and water solubility, its further application in clinical treatment has been severely hindered. Wang et al. developed a cancer cell ablation agent for PDT by using a water-soluble conjugated polymer (PBF) photosensitizer with red light emission [149]. PBF with red emission interacted with negatively charged 3,3′-dithiodipropionic acid disodium salt (SDPA) in an aqueous solution to form uniform nanoparticles. After irradiation with 590 nm light, PBF nanoparticles generated reactive oxygen species (ROS) at the tumor site and rapidly killed cancer cells. In order to achieve precise localization imaging of different parts of the cells and further improve the targeting performance of the material, Feng et al. demonstrated the use of antibody-functionalized CPNPs for targeted cancer cell killing in PDT [150]. CPNPs with yellow fluorescence were prepared by using PFT/polystyrene as the core and surface-modifying PEG-COOH, further modified with an anti-EpCAM antibody and Tat peptide, achieving precise targeting of human MCF-7 cancer cells and photodynamically killing tumor cells. Liu et al. reported a near-infrared fluorescent conjugated polymer with both aggregation-induced luminescence and strong reactive oxygen species generation properties [151]. PTPEAQ was synthesized from a novel polymeric material (TPE) and an anthraquinone (AQ) compound. It has a strong luminescence intensity and ROS formation capability. To achieve targeted and imaging-guided ablation of PDT cancer cells, HER2 was attached to the surface of PTPEAQ-NPs to obtain PTPEAQ-NP-HER2, which can be selectively internalized into SKBR-3 cancer cells to generate the PDT effect to kill cancer cells under white light irradiation. This is the first report of aggregation-induced luminescent conjugated polymers combined with FR/NIR fluorescence imaging for efficient PDT of tumor cells. In addition, PDT systems constructed from CPs can be excited by NIR lasers, which could overcome the light penetration problem and be explored for the diagnosis and treatment of tumors in deeper tissues. Pu et al. reported a mixed-particle doping method used to adjust the NIR photodynamic properties of conjugated polymer nanoparticles to optimize cancer treatment [131]. The treatment system is composed of conjugated polymers that absorb NIR (PCPDTBT) and cerium oxide nanoparticles. They are used as PDT reagents and ROS modulators. Under NIR laser irradiation, ROS production in doped polymer nanoparticles was downregulated under neutral conditions compared to undoped polymer nanoparticles and upregulated under acidic conditions. By adjusting the pH, the photodynamic properties significantly enhanced PDT and significantly reduced non-specific damage to normal tissues in cancer treatment. Luo et al. designed a near-infrared-activated nano-gel particle, NGS@TMZ/ICG [132]. Prulan was crosslinked with poly(deca-4,6-diynedioic acid) (PDDA), an oxidizable and degradable polydiacetylene derivative, and loaded with temozolomide (TMZ) and indocyanine green (ICG), a first-line clinical drug for glioblastoma multiforme (GBM) therapy, to form a nano-gel (NGs@TMZ/ICG) that can be activated under 808 nm laser irradiation. Under irradiation, ICG generates ROS to induce PDDA degradation to small molecules of succinic acid with good biocompatibility, and the nano-gel becomes very soft and releases drugs for GBM treatment in a controllable manner. In order to further prolong circulation in the blood, nano-gels were encapsulated by the erythrocyte membrane and modified with apolipoprotein E (ApoE), which achieved controlled drug release and GBM treatment under 808 nm irradiation. In addition, because the tumor microenvironment usually has insufficient oxygen in PDT due to the imperfect vascular system, the therapeutic effect on solid tumors is limited. To address this issue, Wang et al. constructed PDT systems with photosensitizer-loaded hemoglobin-polymer adducts (HbTcMs), which provide additional oxygen to enhance the therapeutic effect of PDT [152]. Amphiphilic triblock copolymers (mPEG-b-PAA-b-PS) were synthesized by atom transfer radical polymerization, and the photosensitizer TCPP and copolymers were then covalently coupled to hemoglobin (Hb) to form HbTcMs. HbTcMs not only have the same oxygen-binding capacity as Hb, but also exhibit strong antioxidant properties and stability against trypsin digestion. HbTcMs couples produce singlet oxygen and show significant inhibition of 4T1 tumors under in vitro irradiation, exerting better phototoxicity under Hb-supplied oxygen. During photosensitizer-mediated PDT, the need for an external light source hinders the application of PDT in deep-tissue therapy, due to the limited penetration depth. To overcome this problem, Wang et al. developed a novel bioluminescence resonance energy transfer (BRET)-based system for cancer treatment and microbial infection [153]. In this system, cationic oligo-phenylene vinylene (OPV) was used as the photosensitizer, and luciferin, horseradish peroxidase (HRP), and hydrogen peroxide were used as the bioluminescence system. The energy of in situ bioluminescence is absorbed by OPV through the BRET process, sensitizing surrounding oxygen molecules to generate reactive oxygen species (ROS) to kill cancer cells and pathogenic microorganisms in vivo and in vitro. Wang et al. established a novel chemical luminescence resonance energy transfer (CRET) system and used Hb-conjugated CPNPs to prepare an efficient PDT system that can self-luminesce and self-oxygenate [130]. PSMA was co-precipitated with MEH-PPV to synthesize CPNPs, which were then functionalized with Hb (Hb-NPs). To improve the stability of Hb during circulation, Hb-NPs were encapsulated in fusion liposomes (Hb-CPs@liposome). Under the catalysis of Hb, luminescence is absorbed effectively by the prepared Hb-CPs@liposome after reaction with luminol and hydrogen peroxide, sensitizing oxygen provided by hemoglobin to produce ROS and kill cancer cells. This new system provides an effective strategy for PDT by combining chemical luminescence for internal activation and self-oxygenation, overcoming the need for external light sources with limited penetration depth and avoiding molecular oxygen deficiency under hypoxia conditions (Figure 9).

#### 2.2.2. Photothermal Therapy (PTT)

PTT is of great interest in cancer treatment because of its high performance in tumor ablation and minimal side effects [154,155]. In the PTT process, nanomaterials with strong absorbance convert light into heat to ablate cancer cells [156,157,158]. Despite the rapid development of photothermal agents in the last few years, their further clinical application is often limited by low photothermal conversion efficiency (PCE), poor water solubility, and non-biodegradability. To address these difficulties, the photothermal conversion efficiency and water solubility of these conjugated polymers can be improved by introducing some new functional groups. Pu et al. improved the production of non-radiative heat by using an intramolecular orbital engineering strategy [92]. In this approach, electron-withdrawing carbon dots were paired with CPs to facilitate photo-induced electron transfer, thereby increasing photoacoustic brightness and PTT efficiency. Furthermore, Pu et al. proposed another method to enhance photothermal conversion efficiency by incorporating ethylene units into the main chain of conjugated polymers [159]. This design utilizes the enzymatic oxidation properties of ethylene bonds, combined with polymer chemistry, to synthesize a biodegradable semiconductor polymer (DPPV) and transform it into water-soluble nanoparticles (SPNV). The presence of ethylene bonds in the polymer backbone significantly increased the mass absorption coefficient of SPNV by 1.3 times, and the photothermal conversion efficiency was significantly improved by 2.4 times. However, in the PTT process, high PCE is the key to achieving efficient PTT. Liu et al. achieved efficient in vitro and in vivo PTT by utilizing the D-A design of CPs [160]. By introducing intramolecular charge transfer along the main chain of porphyrin-containing conjugated polymers (PorCP) with D-A structure, then encapsulating PorCP and modifying it with Tat, the resulting PorCP-Tat NPs not only reduced aggregation-induced quenching non-radiative decay, but also exhibited strong cell uptake capability, photostability, and a high PCE of 63.8%. Meanwhile, this research provides a new design route for potential personalized therapeutic diagnostic nanomedicine by using conjugated polymer photothermal therapy materials. Wang et al. reported a new type of PTT material based on CPNPs that showed strong NIR absorption and ultra-high PCE in vivo [161]. Firstly, four CPs based on diketopyrrolopyrrolo (DPP) were designed and synthesized with different absorption wavelengths in the near-infrared region. CPNPs were prepared using nanofabrication techniques to improve their water solubility and biological applicability. A PCE of up to 65% can be achieved under NIR laser irradiation. More importantly, the treatment successfully achieved in vitro and in vivo photothermal ablation of cancer cells. Although CPNPs have shown promising prospects as photothermal carriers in the diagnosis and treatment of cancer, they are not ideal photothermal agents. The low level of visualization of PTT formulations, in terms of transportation, distribution, metabolism, and digestion in the body, greatly limits their widespread clinical application. Therefore, the combination of PTT and imaging can help establish a multifunctional diagnosis and treatment system and improve the effectiveness of diagnosis and treatment. Yuan et al. reported a PTT system for PAI guidance in which ultra-small Pdots (4 nm) were synthesized from DPP-BTzTD with a wide spectral response between 800 and 1300 nm, enabling near-infrared light-induced photothermal conversion of NIR-II NPs [129]. The obtained NPs show higher PA signal strength than large-sized NPs, which may be due to a larger specific surface area and better light-capture ability. In addition, compared to larger NPs, they have faster in vivo clearance rates. Therefore, ultra-small NPs are beneficial for efficient cancer cell ablation and rapid excretion from the body without toxicity to mice, and they avoid unnecessary side effects throughout the treatment period. NPs have excellent PCE (53%), enhanced penetration depth, and minimal tissue exposure, providing great prospects for precise diagnosis and treatment of tumors. In terms of PAI-guided PTT, Ding et al. reported a CPN with a D-A structure (PBDT-DIID), which effectively inhibited tumor growth [162]. Guo et al. developed a biodegradable conjugated oligomer nanomaterial for efficient tumor photothermal therapy [163]. The main chain of the conjugated oligomer consists of one strong electron donor (D) and two strong electron acceptors (A). This “A-D-A” structure not only effectively reduces the molecular energy gap, but also enhances the intramolecular charge transfer within the molecule, causing its fluorescence to almost completely quench. The flexible “D-A” structure promotes intramolecular rotation, enhancing the non-radiative transition to generate higher heat. To improve their water solubility, the conjugated oligomers were wrapped with PEG. The photothermal conversion efficiency of resulting nanoparticles (F8-PEG NPs) was up to 82%, making it possible to achieve efficient PAI and PTT. Huang et al. synthesized conjugated polymer nanoprobes (L1057 NPs) with high fluorescence brightness and various excellent properties, enabling real-time imaging monitoring of systemic and cerebral vasculature and clear detection of tumors [128]. L1057 NPs have excellent photothermal characteristics, with a maximum permissible exposure (MPE) limit at 980 nm, making them suitable for tumor PTT under safe laser doses (Figure 10). These results open up new possibilities of using CPs in various integrated diagnostic and treatment strategies based on imaging-guided PTT.

#### 2.2.3. Synergistic Therapy

Currently, in addition to traditional surgery, radiotherapy, and chemotherapy, a series of new tumor treatment methods/strategies have been established, such as biological therapy, immunotherapy, physical therapy (thermal, optical, magnetic, electrical, ultrasound), and dynamic therapy. However, due to the influence of tumor stem cells and complex tumor microenvironment, the difficulty of treatment in the middle and late stages and recurrence have become obstacles to overcoming cancer. Some studies have found that the effect of tumor cooperative treatment is significantly better than that of single treatment. If two or more different methods are combined to treat tumors cooperatively, it will improve the curative effect and decrease side effects [164].

Phototherapy mainly consists of PDT and PTT [165]. In recent years, PDT and PTT have emerged as two representative light-triggered treatment modalities [166]. Under laser irradiation, tumor cells can be effectively killed by the high heat or reactive oxygen species generated by the photosensitizers. Liu et al. constructed a multifunctional CPNP system consisting of two types of CPs [136]. PFVBT has bright red fluorescence and efficient ROS production ability, while PIDTTQ has efficient photothermal conversion and is used as a reagent for PDT/PTT combination therapy through FI/NIR imaging. These two CPs were wrapped in PEG in a CPNPS, and the co-loaded CPNPs showed red fluorescence with a fluorescence quantum yield of 23.1% at 530 nm, a singlet oxygen quantum yield of 60.4%, and a PCE of 47.6% at 808 nm. Subsequently, after surface modification with an anti-HER2 antibody, the obtained CPNPs could target SKBR-3 breast cancer cells and significantly inhibit tumor growth through the synergistic effect of PDT/PTT combination therapy. However, the above-mentioned reports on PDT/PTT synergistic strategies rely on complex construction processes to prepare multi-component drugs. In addition, they require two different wavelengths of lasers for PTT and PDT, which not only lengthens the tumor treatment time, but also weakens the therapeutic effect. Therefore, there is an urgent need to develop effective single-component drugs for PTT/PDT synergistic cancer therapy under single-laser irradiation. Chen et al. developed dual-functional CPNPs using CPs with D-A structures, providing effective energy conversion for NIR light-induced PDT/PTT therapy [167]. The CPs (BIBDF-BT) were synthesized by using alkyl-linked branched thiophene (BT) as the electron donor and (3E,7E)-3,7-bis(2-oxoindolin-3-ylidene)-benzo[1,2-b:4,5-b’]-difuran-2,6(3H,7H)-dione (BIBDF) as the electron acceptor. The CPs were made into CPNPs using PEG-PCL copolymers. CPNPs had significant near-infrared absorption at 782 nm and good anti-photobleaching properties. By inducing charge transfer through D-A system excitation, triggering the transition from singlet state to triplet state, and resulting in non-radiative decay-generating high-thermal energy, favorable reactive oxygen species and effective heat effects were produced at the tumor site (Figure 11). This study provides directions for further developing single-component CPNs for PDT/PTT synergistic tumor therapy. Although PEGylated amphiphilic polymers have been widely used for different components of PDT/PTT synergistic tumor therapy, improving their dispersibility in water, this method often leads to decreased encapsulation and absorption efficiency. Zwitterionic species are a widely used alternative to PEG polymers, which can reduce the size of nanoparticles while avoiding the oxidation of PEG under physiological conditions. Cai et al. further reported a novel zwitterionic CPNP for PAI-guided PDT/PTT synergistic dual-mode therapy [134]. The CPNPs have an absorption wavelength of 700–850 nm, and light at these wavelengths can penetrate deep into tissues. Under 808 nm laser irradiation, the CPNPs produce both ROS and heat in HepG2 cells, leading to cancer cell death. Under the guidance of PAI, after intravenous injection of NPs, PDT/PTT synergistic treatment was confirmed to completely inhibit tumor growth and induce tumor remission, and there was no recurrence. Therefore, CPNPs with PTT/PDT synergistic therapy as a single component can provide efficient and simple enhancement of cancer treatment. Water-soluble CPs were designed by modifying side chains for biologic applications without the need for further encapsulation. Feng et al. used the donor-acceptor (D-A) strategy to design the skeletal structure and reported novel water-soluble PTDBD NPs for near-infrared absorption and PDT/PTT synergistic tumor therapy [168]. The PTDBD NPs were synthesized using electron-rich thiophene and electron-deficient benzo[4,5]thieno[2,3-d] pyrimidine and dione pyrrole-2,5-dione, modified by a cationic quaternary ammonium (QA) group to form hydrophilic PTDBD NPs through self-assembly. The PTDBD NPs strongly absorb in the range of 600–1000 nm, and, under 808 nm laser irradiation, can effectively convert light into heat and, simultaneously, generate ROS. Therefore, PTDBD NPs can kill HeLa cells and eliminate tumors in mice after PDT/PTT synergistic treatment.

In recent years, chemotherapy combined with phototherapy, as a promising means of tumor therapy, has received widespread attention [169,170]. Among them, photothermal chemotherapy combined with antineoplastic drugs and targeted killing of tumor cells can not only effectively solve the problem of multidrug resistance in tumor cells, but also have a synergistic effect and improve the prognosis of tumors [171,172]. Compared with the single-treatment model, the compound system is expected to reduce the recurrence rate of tumors and the side effects. However, there are many kinds of nanocomponents in the system, which can easily lead to the instability of the nanostructure and the contradiction between the components. Therefore, Pu et al. designed a diagnostic nanotherapeutic system (DCPN), with few components and a simple structure, to enable photothermal treatment of chemotherapy guided by NIR FL/PA imaging [173]. Using an amphiphilic semiconductor polymer (PEG-PCB) as a multifunctional nanocarrier, PEG-PCB with a hydrophobic-conjugated backbone interacts with the anti-cancer drug DOX to obtain drug-carrying nanoparticles (DCPNs). The nanoparticles have only a two-component, multimodal therapeutic diagnostic capability. DCPNs with the highest drug-loading capacity in PAI provided 3.8-fold higher amplitude in the tumor region, as well as excellent anti-tumor efficacy in synergy with chemotherapy and PTT. Li et al. reported a novel amphiphilic conjugated polymer with side chain PEG units as a nanocarrier that exhibited maximum absorption in the ideal photothermal treatment window between 800 and 850 nm [174]. Without the assistance of other PEG polymers, chemical drugs (Dox) can be directly encapsulated into self-assembled PEGylated poly(diketopyrrolopyrrole-thiophene) (PDPPT) nanoparticles. Under 808 nm laser irradiation, they exhibit a 76% photothermal conversion efficiency while providing synergistic treatment with chemotherapy and PTT under the monitoring of photoacoustic imaging. However, the uncontrolled release of small-molecule drugs, delivered together with semiconductor polymers, severely limits their clinical application and inevitably leads to systemic toxicity. Researchers hope that controlled release of drugs can reduce drug loss during delivery and enable rapid release of drugs in tumor cells. Due to the non-specific distribution and rapid clearance of chemotherapy drugs, it is necessary to develop more specific and effective delivery methods for therapeutic agents. Zhang et al. developed redox-responsive polymer vesicles using amphiphilic triblock copolymer (PCL-ss-PEG-ss-PCL)-developed redox responsive polymer vesicles [175]. In order to avoid the limited efficacy of chemotherapy drugs due to systemic exposure and drug resistance, redox-sensitive polymer vesicles carry two types of chemotherapy drugs, Dox and paclitaxel (PTX), and also encapsulate indocyanine green (ICG) for photothermal therapy. Penetrating peptides and LHRH-targeting molecules are modified on the surface of polymer vesicles. The results indicate that drug release is triggered in a reductive environment, high cell uptake is achieved through dipeptide and laser irradiation, and higher cytotoxicity is achieved through chemical photothermal therapy. Therefore, redox-responsive LHRH/TAT dipeptide-PTX/DOX/ICG co-loaded polymer micelles have shown great potential in tumor targeting and chemical photothermal therapy. This drug delivery system provides a powerful strategy for the delivery, targeting, and penetration of multiple types of drugs and achieves high anticancer effects through chemical photothermal combination therapy. In another study, a nanoplatform controlling drug release properties was established to achieve precise treatment of tumors. Wang et al. constructed a pH/NIR light-controlled drug release nanoplatform. By encapsulating a pH/NIR-responsive polymer composed of the photothermal effect DPP semiconductor polymer (PDPP3T) and polystyrene-b-poly(N-isopropylacrylamide-co-acrylic acid) (PSNiAA), polyacrylic acid (PAA) was used as an “intelligent” unit for Dox release in response to pH changes [137]. External light stimulation can also be converted into thermal energy, causing the PSNiAA layer to contract and release drugs on demand under acidic conditions. Thanks to the excellent photothermal properties of PDPP3T nanoparticles and the pH and thermal responsiveness of PSNiAA, PDPP3T@PSNiAA-Dox nanoparticles guided by PAI can more accurately and effectively eliminate tumors through synergistic chemotherapy and photothermal therapy, while minimizing unnecessary drug release. In particular, drug release was enhanced by near-infrared light irradiation under acidic conditions, providing a promising PAI-guided and on-demand chemical-photothermal synergistic therapy for cancer treatment. Cheng et al. proposed to improve the performance of semiconductor polymers by engineering the main chain and alkyl side chains, and then self-assembling the semiconductor polymers with pH-sensitive prodrugs into nanoparticles for PAI-guided chemo-photothermal therapy. By preparing the semiconductor polymer together with the polymer PADD, PADD@SPs was obtained [176]. The synthesized nanoparticles contain pH-sensitive prodrug copolymers that release Dox in acidic environments. In vitro photothermal studies have shown that the nanoparticles can exhibit high PCE (45%) at relatively low NIR power. In vivo antitumor experiments showed that mouse tumor volumes were reduced by about 50% after NIR radiation therapy, with a significantly increased survival rate and survival time. In summary, multifunctional nanoparticles combining photothermal materials and chemotherapy drugs show great potential in anti-tumor therapy.

Compared with other conventional therapies, photodynamic therapy (PDT) has unique advantages, such as high efficiency, minimal invasiveness, no drug resistance, local light controllability, and the tolerability of repeated doses, which have attracted it more attention in cancer treatment. However, the continuous intracellular O2 consumption during the PDT process will generate acute hypoxic microenvironments, which greatly limits the therapeutic effect of PDT. In order to change the adverse effects of hypoxia and improve the therapeutic effect of PDT, the combination of PDT and a low oxygen microenvironment-responsive drug delivery system can transform the adverse effects of hypoxia into positive factors, which helps to improve the efficacy of PDT. Liu et al. constructed a novel azobenzene (AZO) nanocarrier (CPs-CPT-Ce6 NPs) for PDT and chemotherapy combinations [177]. They could co-deliver photosensitizers and chemotherapy drugs for effective PDT and subsequent low oxygen-responsive drug release for chemotherapy. Hypoxia-responsive conjugated polymer chains (AZO-CPs) were synthesized through the condensation reaction of 4,4’-azodiphenylamine-containing azo groups with p-phenylenediformaldehyde, then AZO-CPs were mixed with chemotherapy drug (CPT) and photosensitizer (Ce6) in polyvinylpyrrolidone (PVP) solution, and nanocarriers (CPs-CPT-Ce6 NPs) were prepared by the precipitation method. Under laser irradiation, CPs-CPT-Ce6 nanoparticles can convert oxygen into singlet oxygen, killing cancer cells. The activation of azoreductase under low oxygen conditions can lead to the reduction and cleavage of azo groups. Afterwards, the azo groups in CPs were reduced and cleaved to promote the dissociation of CPs, resulting in the release of chemotherapy drugs, and further killing PDT-resistant cancer cells. Therefore, this strategy can synergistically enhance the anticancer efficiency of traditional PDT. In the process of chemotherapy/PDT combination therapy, due to the nonspecific interactions between drugs and the environment before reaching the target, premature drug release often occurs in the circulation, lacking active targeting of tumor tissues or cells. In order to overcome these drawbacks, stimulus-responsive intelligent nanodrug carriers provide a promising platform for achieving controlled drug release and improving the efficacy of tumor treatments. Tang et al. developed a targeted drug delivery nanocarrier, iRGD-BDox@CPNs. In this system, conjugated polyfluorene-vinylene (PFV), the prodrug BDox (doxorubicin modified with a phenylboronic acid ester group), and internalized RGD-modified amphiphilic polymer (DSPE-PEG-iRGD) self-assembled into nanoparticles iRGD-BDox@CPNs [137]. The PFV core loaded with BDox can efficiently generate large amounts of ROS and achieving efficient drug release and chemotherapy/PDT combination therapy. The RGD peptide can enhance permeability and tumor-targeting ability, and the amphiphilic polymer DSPE-PEG shell can prolong the retention time of nanoparticles in circulation. The biocompatibility of conjugated polymers is good, and they have bright fluorescence for cell imaging and visualization monitoring of drug release. Compared with unmodified Dox, BDox has lower toxicity and higher hydrophobicity, which can improve the stability of the drug and avoid premature leakage of the drug. Therefore, this photo-triggered ROS-responsive drug delivery system based on conjugated polymers achieves controlled drug release, active targeting of tumor cells, and effective chemotherapy/PDT combination therapy, providing a promising strategy for tumor treatment. However, excessive cytotoxic ROS generated by PDT may lead to side effects, such as non-specific damage to adjacent normal tissues and the induction of normal tissue vascular stasis. Therefore, finding a relatively mild PDT (mPDT) method that can effectively reverse multidrug resistance (MDR) through coordinated chemotherapy without causing tissue damage has important practical value. Jin et al. reported a bifunctional drug delivery system based on Ce6, Dox, and poly (phosphorylcholine) nanomicelles that can reverse MDR by combining mPDT with chemotherapy [140]. Firstly, a PDT-responsive nanomicelle was synthesized, which encapsulates the photosensitizer Ce6 and Dox-conjugated polymer micelle (Ce6/pMPC-Dox). Under relatively low near-infrared laser irradiation, Ce6/pMPC-Dox nanomicelles produce a lower concentration of ROS, which is sufficient in quantity and function to effectively inhibit the expression of drug-resistant cells and make them sensitive to chemical drugs without causing further damage to important organs. In addition, further experimental results confirmed that the combination of mPDT and chemotherapy can safely and effectively eradicate multiple drug-resistant tumors in vivo. With the improvement of the requirements for the safety of cancer treatment, the cooperative treatment strategy will show great advantages in reversing multiple drug resistance in cancer treatment, especially for some cancer treatments with high safety requirements. In addition, Xiao et al. reported a self-sacrificially biodegradable pseudo semiconductor polymer (PSP) for NIR-II fluorescence imaging, photodynamic immunotherapy, and photo-activated chemotherapy (PACT) [138]. PSP is known as “self-degradable” and can be triggered by intracellular glutathione (GSH) degradation, which is different from the current non-degradable semiconducting polymers. A reduction-sensitive pseudo-conjugated polymer PSP was designed and mixed with polymer PE^Dox^-containing Dox prodrug to form a nano-drug. NP@PE^Dox^/PSP generates a large amount of ROS under 808 nm light irradiation to achieve PDT and tumor immunotherapy. Subsequently, the generated ROS can disrupt the thione bonds in the polymer, release Dox, and achieve photoactivation chemotherapy (Figure 12). NP@PE^Dox^/PSP emits strong NIR-II fluorescence under light irradiation for biological imaging. More importantly, intracellular glutathione can degrade PE^Dox^ and PSP. In short, NP@PE^Dox^/PSP has superior NIR-II fluorescence imaging ability in the body, which can effectively inhibit the proliferation of cancer cells under light and inhibit tumor growth.

#### 2.2.4. Future Directions for Therapeutics

Very great progress and achievements have been made regarding the therapeutic application of CPs in cancer. However, as most of the therapeutic systems are now undergoing preliminary experiments in vivo, in order to further enhance translational clinical applications, the metabolism and biodistribution of CPs need to be investigated and used to enhance the therapeutic effects of oncology. In addition, new therapeutic platforms based on CPs need to be designed, new strategies for manufacturing drug components need to be constructed, and site-specific release of therapeutic drugs needs to be enabled by remote control technologies. The development of CPs carriers should focus on improving in vivo delivery efficiency and targeting capabilities in order to achieve precise drug delivery to tumor sites and improve drug delivery efficiency. By combining phototherapy with other therapeutic approaches, such as chemotherapy or immunotherapy, CPs may be able to produce extraordinary therapeutic results with the help of synergistic effects.

## 3. Conclusions and Outlook

In this review, we present the progress and achievements of recent research on the design and application of CPs in bioimaging and tumor therapy. We summarized various multicolor imaging strategies to simultaneously study multiple targets, near-infrared imaging, and two-photon imaging utilizing FRET to achieve deep-tissue and super-resolution imaging to visualize the subcellular world, and photoacoustic imaging to guide cancer diagnosis and therapy. Additionally, for cancer therapy, we reviewed new strategies in recent years based on various therapies produced by light irradiation. CPs have emerged as a new trend in cancer therapy, due to their excellent optical and simple molecular structure design and controllable chemical properties, to achieve multi-function integration, promising to overcome the challenges in tumor diagnosis and treatment.

Although CPs have made significant progress in biological imaging and tumor treatment and have had a major impact on the field of tumor diagnosis, the development of CPs in biomedical applications is still in its early stages, and there are still many areas worth exploring. Therefore, we will focus on future directions related to biological imaging and tumor treatment, hoping to promote the development and progress of fields such as biomedicine. Various probes with excellent performance based on CPs as imaging agents have been developed, and it is necessary to further clarify the metabolic situation after CPs enter the body and conduct long-term toxicity research in vivo, which is an important direction for the future development of biological imaging detection and tumor treatment and is expected to provide powerful evidence for clinical application. At the same time, it is necessary to continue to develop new conjugated polymers and optimize various synthesis methods of nanomaterials, realizing large-scale, green, and low-cost preparation while retaining the photophysical properties of functional groups and main chain structures, to adapt to more complex biochemical environments and play a role in more deep biological systems. It is also necessary to design new structures with longer wavelength absorption and emission bands to promote the penetration depth of light in vivo. Although CPs have been widely used in both in vitro and in vivo biological imaging, current biological imaging mainly focuses on passive imaging, or passive targeting, and the future development direction of targeting imaging still needs further exploration. In addition, it is necessary to strengthen the development of multifunctional CPs, multimodal imaging, diagnostic systems, and comprehensive treatment platforms.

## Figures and Tables

**Figure 1 molecules-28-05091-f001:**
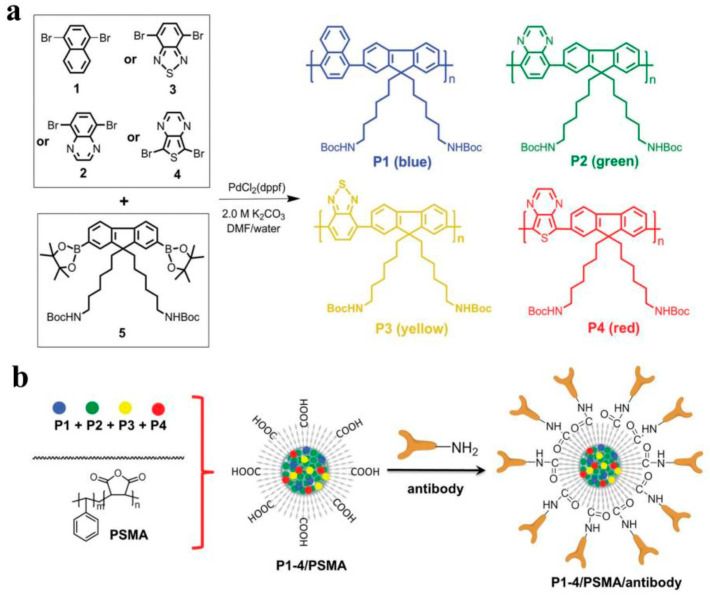
(**a**) Chemical structure and synthesis route of CPs. (**b**) Preparation of polychromatic CPNPs and antibody functionalization modification. Four conjugated polymers with blue light, green light, yellow light, and red light emission were designed and synthesized. Based on the hydrophobic interaction between CPs and poly(styrene maleic anhydride copolymer) (PSMA), they were used to prepare carboxyl functionalized carbon nanotubes by the coprecipitation method [59]. Copyright 2014, Wiley-VCH.

**Figure 2 molecules-28-05091-f002:**
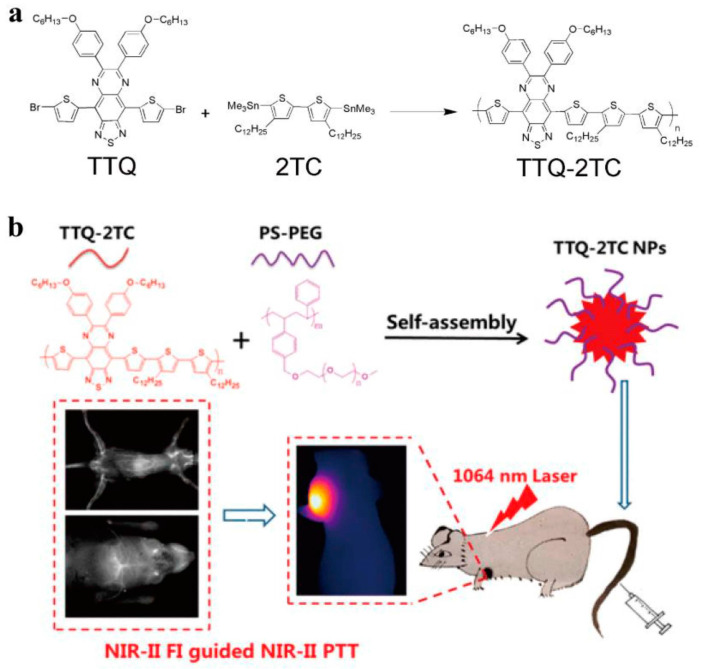
(**a**) Preparation and route synthesis of CPs. (**b**) Preparation of CPs nanoparticles and PTT of tumor site in mice by FI under 1064 nm laser irradiation [73]. Copyright 2020, American Chemical Society.

**Figure 3 molecules-28-05091-f003:**
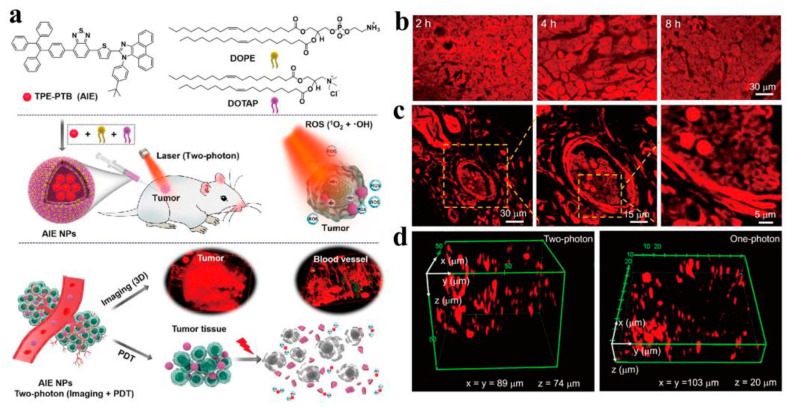
(**a**) The structural composition and preparation method of nanoparticles, and the use of two-photon laser irradiation on mouse tumor sites to guide the application of PDT. (**b**) TPFI of tumor tissue at different times after injection of AIE NP into mice. (**c**) After 4 h of injection of AIE NP, the tumor was removed for imaging. The yellow part marked in the figure is for zooming in and observing the subsequent images. (**d**) Comparison of 3D reconstruction images of tumor tissue under two-photon and single-photon in vitro conditions [80]. Copyright 2020, American Chemical Society.

**Figure 4 molecules-28-05091-f004:**
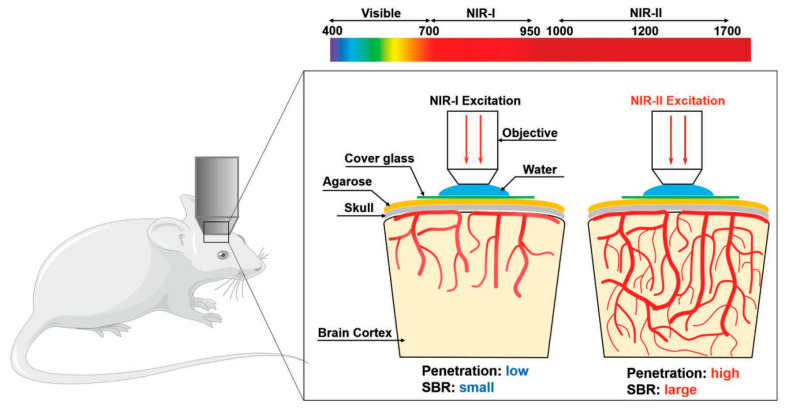
Schematic diagram of the principle of in vivo 2PF imaging of mouse brain stimulated by NIR [81]. Copyright 2019, WILEY-VCH Verlag GmbH & Co. KGaA.

**Figure 5 molecules-28-05091-f005:**
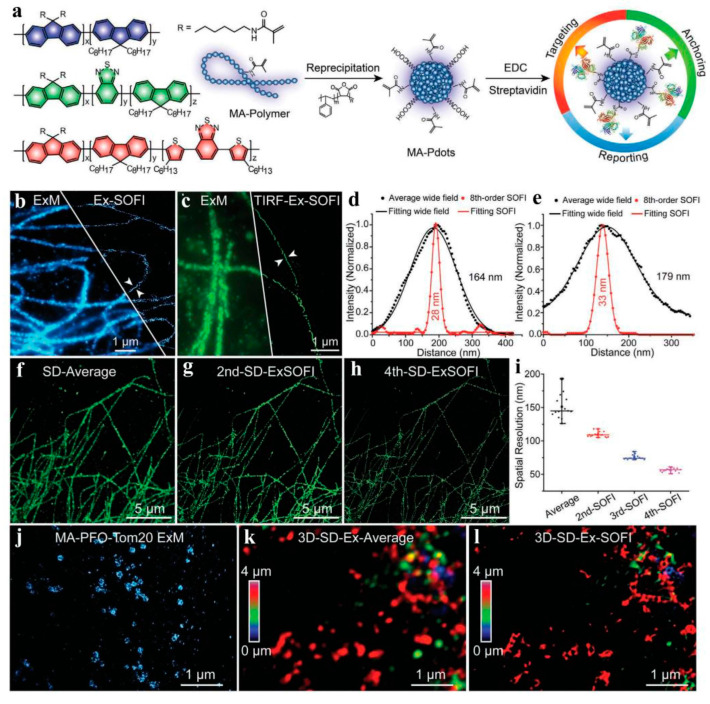
(**a**) Multifunctional nanoparticles (MA Pdots) were synthesized from the methyl propionamide (MA) group. After binding with Streptavidin, MA Pdots carried out molecular targeting and fluorescence reporting simultaneously in the expansion microscope (ExM). (**b**) Under EXM, the TIRF images and eighth-order SOFI images generated by microtubules with MA-PFO Pdots were labeled. (**c**) Under EXM, the TIRF images and eighth-order SOFI images generated by microtubules with MA-PFBT-Pdots were labeled. (**d**,**e**) The intensity distribution between the white arrows in (**b**,**c**), respectively. (**f**) SD confocal images of expanded microtubules in BS-C-1 cells labeled with MA-PFBT. (**g**,**h**) Second-order and fourth-order SOFI images were reconstructed by analyzing the data obtained in (**f**). (**i**) Enhanced Ex-SOFI resolution of microtubules in (**f**). (**j**) CLSM captures ExM images of labeled mitochondrial outer membrane proteins. (**k**) The average SD image of mitochondrial outer membrane proteins labeled with MA-PFBT Pdots and the corresponding fourth-order SOFI image (**l**) [88]. Copyright 2021, Wiley-VCH GmbH.

**Figure 6 molecules-28-05091-f006:**
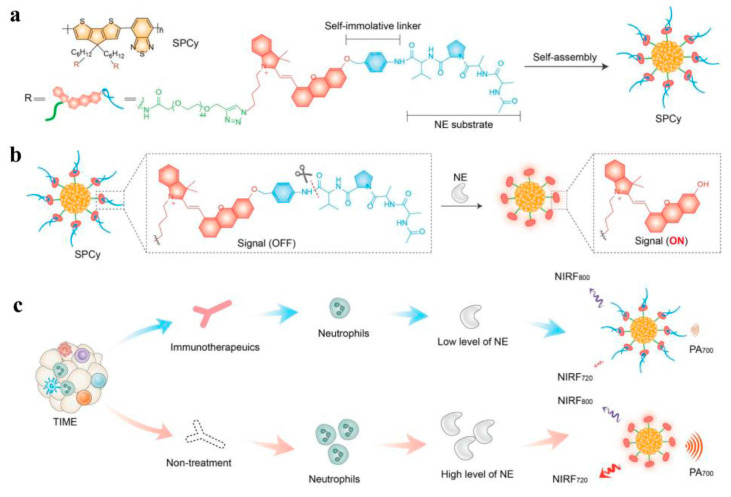
(**a**) The structure of the polymer and the synthesis diagram of SPCy. (**b**) The influence of NE on the structure and signal of SPCy. (**c**) Schematic diagram of imaging mechanism of SPCy [94]. Copyright 2022, Wiley-VCH GmbH.

**Figure 7 molecules-28-05091-f007:**
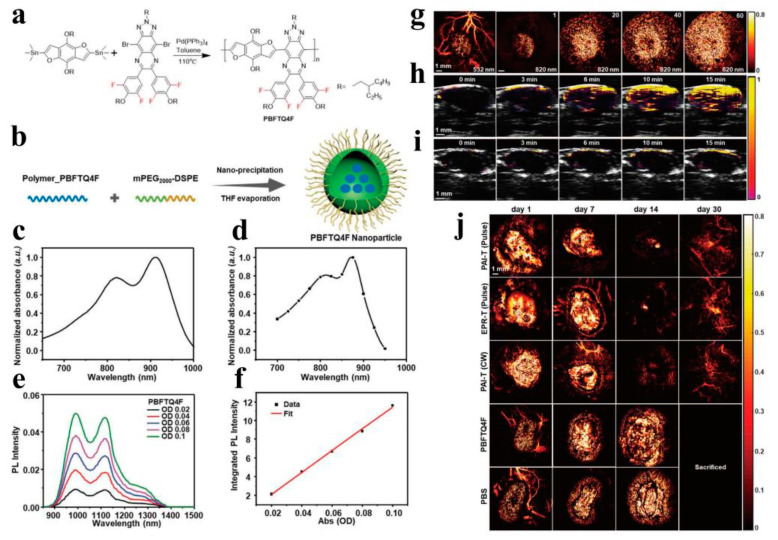
(**a**) The chemical composition, structure, and synthesis process of PBFTQ4F. (**b**) Polymer PBFTQ4F reacted with mPEG-DSPE to prepare nanoparticles. (**c**) Ultraviolet visible absorption spectra of PBFTQ4F NPs solution. (**d**) Photoacoustic absorption spectra of PBFTQ4F NPs solution. (**e**) Emission spectra of PBFTQ4F NPs solution at excitation wavelength of 808 nm. (**f**) The integrated fluorescence intensity plotted at 808 nm for PBFTQ4F NPs solution. (**g**) OR-PAM images of tumors after intravenous injection of PBFTQ4F NPs. Under 532 nm laser, it can be used for vascular imaging. The accumulation of nanoparticles in the imaging area were tracked at different scanning times under 820 nm laser. The number in the upper right represents the number of scans. (**h**) During the simulation of PTT process, AR-PAM and ultrasound images were performed on the distribution of nanoparticles within the tumor at different time points. (**i**) AR-PAM and ultrasound imaging results of the distribution of nanoparticles in tumors at different time points under 808 nm laser irradiation. (**j**) In vivo OR-PAM imaging of tumor mice in each experimental group on the 1st, 7th, 14th, and 30th days after treatment [96]. Copyright 2021, Wiley-VCH GmbH.

**Figure 8 molecules-28-05091-f008:**
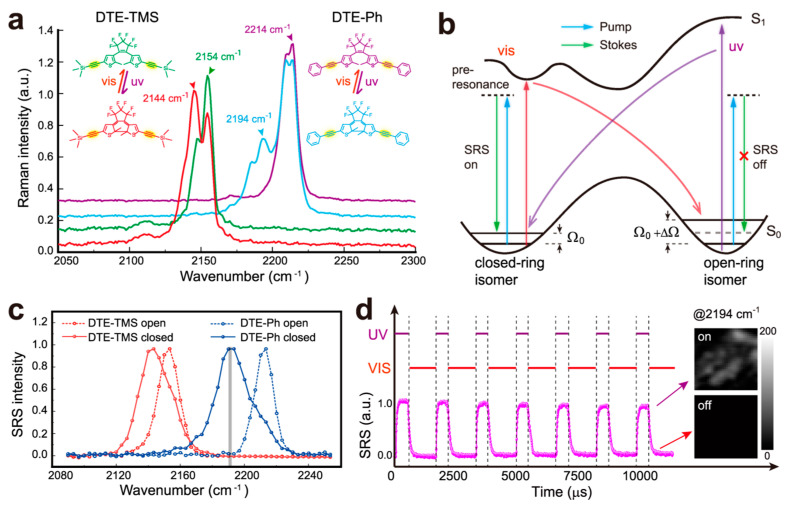
(**a**) Under visible light induction at 360 nm, photocyclization converts the open-ring isomer (corresponding to the green and purple curves in the figure) into a closed−loop structure (corresponding to the red and blue curves in the figure), resulting in a red shift in the spontaneous Raman spectrum of alkynes. Under visible light irradiation at 633 nm, the reverse process of photocyclization reversal of ring−opening isomers and the restored Raman spectrum are generated. The partial photocyclization under ultraviolet radiation may be caused by the two-photon effect pumped by 1064 nm Raman. (**b**) Schematic diagram of SRS conversion related to photoisomerization under fixed detection frequency of closed-loop isomers. (**c**) RS spectra of DTE−TMS and DTE−Ph in open−loop and closed−loop forms. (**d**) The SRS signal of DTE−Ph obtained at a UV induced Raman frequency (2194 cm^−1^) shows the switching behavior under UV/visible light pulse irradiation [103]. Copyright 2021, The Author(s).

**Figure 9 molecules-28-05091-f009:**
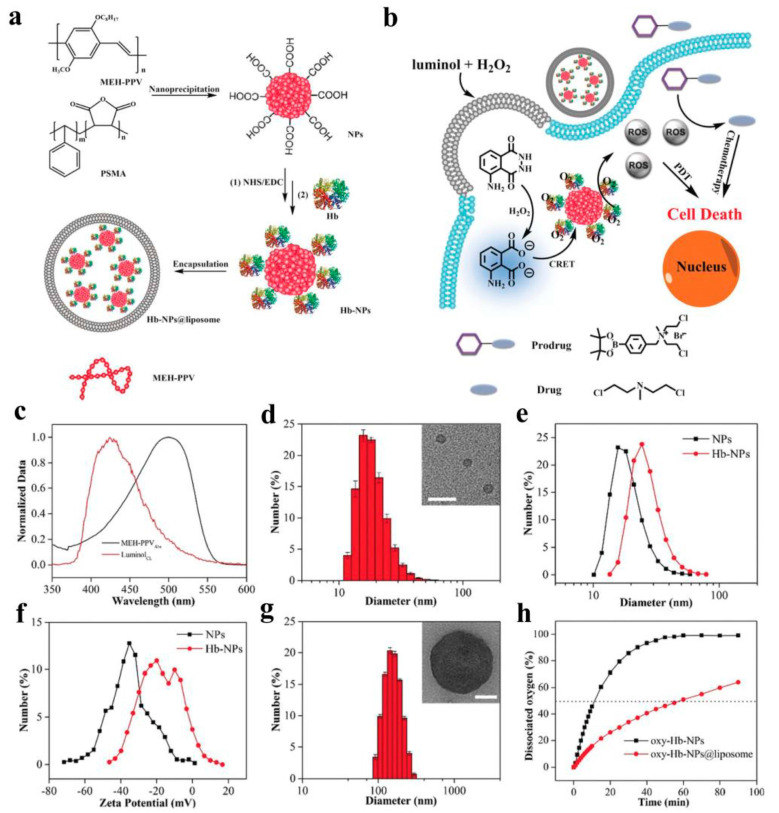
(**a**) Schematic diagram of Hb−NPs@liposome preparation. (**b**) PDT therapy based on bioluminescence resonance energy transfer system. (**c**) The UV−visible absorption spectrum and luminol luminescence spectrum of MEH−PPV. (**d**) Size distribution of MEH-PPV NPs; The illustration shows TEM images of MEH−PPV NPs. (**e**,**f**) Hydrodynamic diameter and potential changes before and after the connection between nanoparticles and Hb. (**g**) Hb−NPs@liposome size distribution; The illustration shows the typical EM image of Hb−NPs@liposomes. (**h**) Oxygen dissociation curves of oxy Hb−NPs@liposomes and oxy Hb−NPs carrying oxygen [130]. Copyright 2019, Wiley-VCH Verlag GmbH.

**Figure 10 molecules-28-05091-f010:**
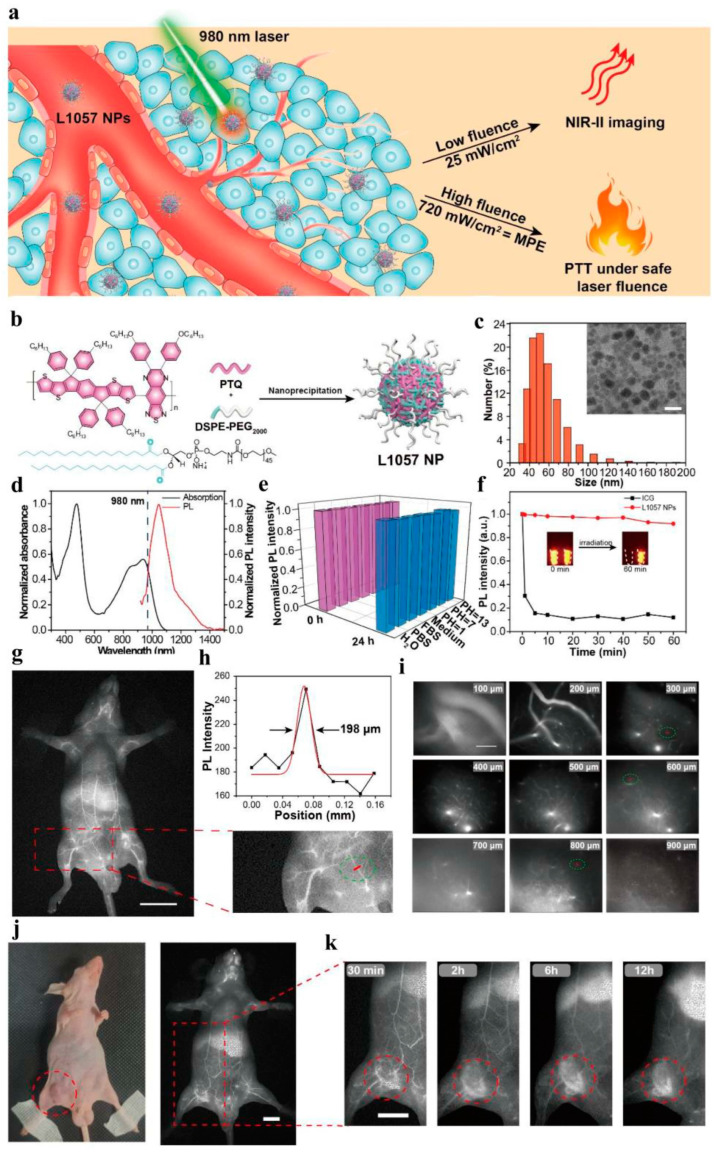
(**a**) Schematic diagram of the principle of using L1057NPs as therapeutic drugs to enter the body and perform PTT on tumor sites under NIR II guidance. (**b**) The chemical composition and structure of various polymers, as well as the preparation process of L1057 NPs. (**c**) Size distribution and TEM diagram of L1057NPs. (**d**) The UV-visible absorption and emission spectra of L1057NPs. (**e**) Comparative images of fluorescence stability in different solutions of L1057 NPs. (**f**) Comparative image of photostability of L1057 NPs and ICG in water under 808 nm laser irradiation. (**g**) Whole-body imaging of mice treated with L1057 NPs under 980 nm laser excitation. (**h**) The red line in the figure refers to the fluorescence intensity distribution diagram of the cross-section (circled in green in the figure). The red curve is the Gaussian function of data fitting. (**i**) After intravenous injection of L1057 NPs, at different depths (100-900 μ m) Brain vascular imaging. (**j**) Nude mouse carrying breast tumors. (**k**) At different times, L1057 NPs were used to treat subcutaneous 4T1 breast tumor mice with NIR-II fluorescence imaging [128]. Copyright 2020, American Chemical Society.

**Figure 11 molecules-28-05091-f011:**
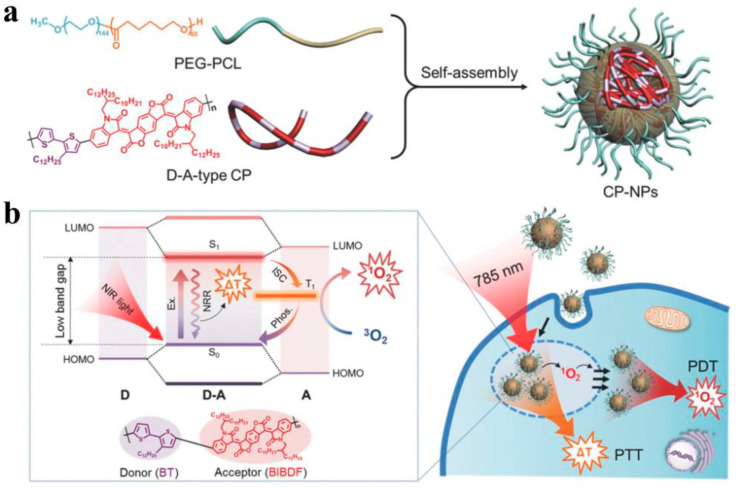
(**a**,**b**) Schematic diagram of nanoparticles and the photophysical mechanism of photosensitive cancer treatment [167]. Copyright 2017, Wiley-VCH Verlag GmbH.

**Figure 12 molecules-28-05091-f012:**
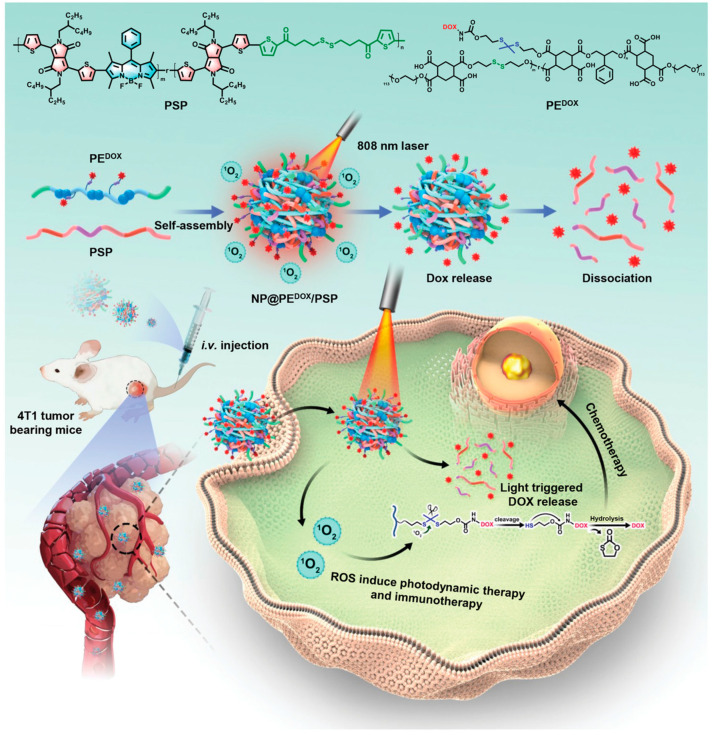
Schematic diagram showing the design of self-sacrificial degradation NIR-II therapeutic instrument NP@PE^Dox^/PSP, used for photodynamic immunotherapy and PACT [138]. Copyright 2022, Wiley-VCH GmbH.

**Table 1 molecules-28-05091-t001:** Advantages and applications of various imaging techniques.

Title	Advantage	Application	Ref.
Multicolor imaging	Being able to observe multiple cell structures simultaneously at the same excitation wavelength provides more important physiological results and greatly solves the problem of spectral overlap caused by multiple fluorescence.	Application of tumor cell imaging and detection. Multiple biological analysis and diagnostic applications. Selective internalization and imaging of malignant lysosomes, as well as real-time tracking, 3D, and polychromatic cell imaging applications.	[60,61,104,105,106]
Near-infrared imaging	Has good sensitivity, spatiotemporal resolution, high signal-to-noise ratio, and easy operation.	Real-time cell tracking application in vivo. In vivo blood and tumor imaging. In vivo cytotoxicity analysis and microvascular imaging. In vivo, deep-tissue, and ultrafast imaging of mouse arterial blood flow. Non-invasive cranial brain imaging.	[69,71,72,73,107]
Two-photon imaging	The scattering coefficient in biological tissues is small, with good penetrability, relatively small damage to biological tissues, and low phototoxicity.	Used for 3D reconstruction of intact mouse brain and skull cerebral vascular network. Two-photon fluorescence imaging of cells, tissues, and organs in living animals. Cell imaging, in vitro tissue imaging, and vascular imaging. Deep high-resolution imaging of microcracks in bone.	[78,81,82,108,109]
Super-resolution imaging	It can clearly observe cellular structure and subcellular structure, which overcomes the optical diffraction limitations of optical microscopy, and has good temporal and spatial resolution.	Super-resolution long-term visualization for subcellular dynamics. Used for in vivo brain imaging. In vivo imaging of animal brain microvessels.	[86,87,88,110,111]
Photoacoustic imaging	PAI signals can penetrate deeper tissues, are non-invasive, non-radiative, have high imaging resolution, good contrast, strong sensitivity, and can provide multi-scale and multi-dimensional image information.	PAI of whole-body lymph nodes in mice. Used for in vivo imaging and treatment of tumors. Real-time imaging and optical urine analysis. Whole-brain photoacoustic imaging in animal models. Imaging of mouse brain and cerebral blood vessels.	[93,94,96,112,113]
Raman imaging	High specificity, high sensitivity, fast scanning speed, can avoid self-luminescence problems, low background signal, high spatial resolution, high chemical specificity, multiplexing ability, excellent optical stability, and non-invasive detection ability.	Raman imaging of cells and tissues. Used for non-invasive microvascular imaging in vivo.	[101,102,103,114]

**Table 2 molecules-28-05091-t002:** Schematic diagram and application of tumor treatment.

Title	Schematic Diagram	Application	Ref.
PTT	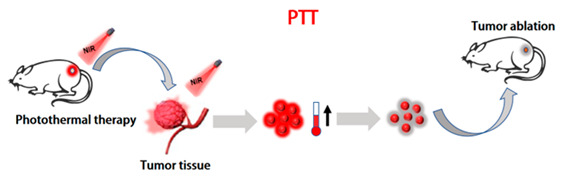	PTT has the advantages of small adverse reactions and high specificity. Combined with a variety of imaging, it can achieve visual and effective local killing of tumors. Usually used to treat breast cancer and prostate cancer.	[126,127,128,129]
PDT	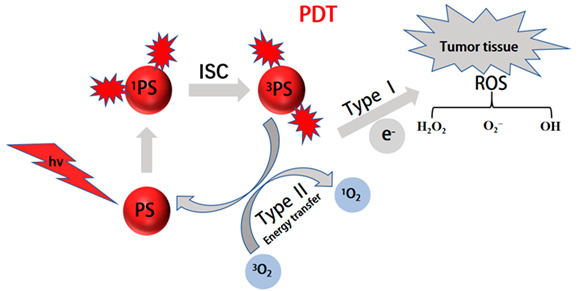	PDT therapy has the advantages of good selectivity, minimal trauma, low toxicity, and good applicability, which can protect the functions of tissues and important organs. Can effectively resist bacteria and treat tumors.	[130,131,132,133]
PTT-PDT	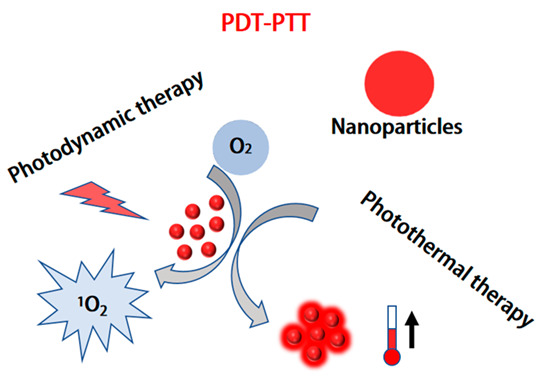	During the combined application of PTT and PDT, PTT enhances CAT activity, promotes an increase in oxygen content, alleviates hypoxia, and improves PDT. The free radicals generated by PDT disrupt the expression of heat shock proteins, thereby improving PTT. From this, PDT and PTT mutually promote and synergistically improve the anti-tumor effect. The multimodal therapy, combined with PTT and PDT, has broad prospects in combating multiple-drug resistance (MDR) and hypoxia-related tumor resistance.	[134,135,136]
PTT-Chem	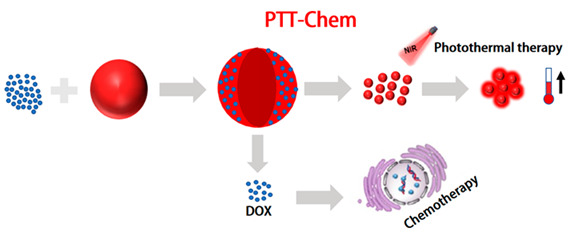	Nanomaterials loaded with chemotherapy drugs can passively target tumors by enhancing penetration and retention effects, or actively target tumors by surface-binding molecules. Local heating during photothermal therapy can also improve cell membrane permeability and drug cytotoxicity, achieving a “1 + 1 > 2” therapeutic effect and inhibiting tumor recurrence.	[126,137]
PDT-Chem	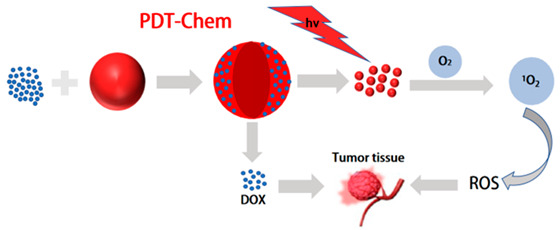	The synergistic effect of PDT chemotherapy solves the limitations of insufficient local drug concentration and severe adverse reactions during chemotherapy, overcomes tumor resistance and increases anticancer activity, and treats tumors by exerting synergistic effects.	[138,139,140]

## Data Availability

The data presented in this study are available in “Conjugated Polymeric Materials in Biological Imaging and Cancer Therapy”.

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
