# Peer review of "Conjugated Polymeric Materials in Biological Imaging and Cancer Therapy"

_molecules, 2023, doi:10.3390/molecules28135091_

Round 1

Reviewer 1 Report

Current manuscript describes applications of conjugated polymeric in biomedicine especially bioimaging and cancer therapy. I think it is interesting for readers and can be published after revision.

1.Authors must add the following sections for better description of CPs. I suggest to authors read the articles; https://doi.org/10.1021/acs.jmedchem.9b00803; https://doi.org/10.1016/j.apmt.2021.101117

- Chemical and electrochemical synthesis of CPs

-Physicochemical and biological properties of CPs

2. I recommend authors add a comprehensive table regarding applications of CPs in biomedicine. This can be useful for readers.

3. The CPs have limitations in clinical experiments. Please discuss limitations of CPs in bioimaging and cancer therapy.

4. Please add a section concerning future perspective of CPs in bioimaging and cancer therapy.

5. Please elaborate captions in the figures 1,2,3, and 5.

6. Please add abbreviations of all figures at captions.

7. Authors must discuss results reported by researchers, not just report. For example: "Wang et al. synthesized a novel water-soluble conjugated polymer (PBF) with red light emission, which interacts with negatively charged disodium 3,3’-dithiodipropi- onate (SDPA) in aqueous solution through electrostatic interaction to form nanoparticles with uniform size[117]. After irradiation with a specific wavelength of light, PBF nanoparticles produce cytotoxicity at the tumor site and quickly kill cancer cells. In order to further enhance the targeting performance of the material."

No comment.

Reviewer 2 Report

Please consider the comments point-by-point for the manuscript “Conjugated polymeric materials in biological imaging and cancer therapy”:

1-      Why did you opted conjugated polymeric materials for this study?

2-      Please present table about advantages and disadvantages of  nanomaterials for cancer therapy. The below references may be helpful:

·         Self-assembled nanostructures for anticancer applications: Advances and limitations (https://www.nmb-journal.com/article_161602.html)

·         Various novel strategies for functionalization of gold and silver nanoparticles to hinder drug-resistant bacteria and cancer cells (https://www.mnba-journal.com/article_152629.html)

3-      At the end of the introduction, please explain innovative aspects of your study compared to previous studies.

4-      Figures have not suitable quality. Please improve quality of all figures, specifically figures 9, 10, and 11.   

5-      The discussion for anticancer mechanism of ROS is weak and should be clarified by related references. the below reference may be helpful:

·         ROS and RNS modulation: the main antimicrobial, anticancer, antidiabetic, and antineurodegenerative mechanisms of metal or metal oxide nanoparticles (https://www.nmb-journal.com/article_167527.html)

6-      The conclusion section can be reorganized based on the main findings of the study.

7-      The manuscript would benefit from English grammar editing.

the manuscript would benefit from English grammar checking by a native person.

Reviewer 3 Report

This article by Zheng et al. provides a comprehensive review of the significant research progress made in the application of conjugated polymers (CPs) in multiple fields, including chemistry, medicine, life science, and material science. The authors specifically discuss the latest research on CPs in cell imaging, tumor diagnosis, and treatment and highlight future development directions in these areas. While the article is important and merits publication, there are several issues that need to be addressed before publication. The following are some comments that should be considered before finalizing the manuscript.

General Comments:

·         The authors seem to have included numerous reports in each section, but without providing sufficient details. Additionally, their approach appears to be merely discussing individual articles one after another, without weaving them together into a coherent narrative that illustrates the field's progress or how an article has overcome the limitations of a previous one. To improve the paper, the authors could either offer more comprehensive explanations or eliminate some of the less relevant reports. One way to accomplish this would be to use tables to provide a concise summary of the articles under discussion.

·         The article may benefit from a comprehensive review as there are several sentences that are difficult to comprehend. This would help to ensure that the article is free from grammatical errors and able to communicate its ideas effectively.

Specific Comments

1.       It would be beneficial to clarify in the introduction why this review article is important and its significance.

2.       Consider discussing previous articles and highlighting how this review is different from similar ones.

3.       Section 2.1 is mentioned twice, please correct this repetition.

4.       Before starting with the subsections in P3 L112, it may be helpful to give a brief overview of what the authors are going to discuss in the next subsections and how they are divided.

5.       In Section 2, the authors briefly discuss many different previous reports, which can become challenging to follow after a while. Consider summarizing the results in the form of a table to help readers easily understand this section.

6.       Section 2 is too long, and it may be helpful to divide it into smaller subsections, if possible.

7.       Please explain the caption of Fig. 1 further, including the name of the polymer and a brief explanation of the figure in the caption. Also, mention what a, b, and c represent. A similar issue can be found in Fig. 5 and 8.

8.       When ending a subsection (e.g., 2.2.1 or 2.2.2), consider adding some concluding remarks about the contents in the section, rather than ending it abruptly.

9.       Consider preparing a table summarizing all four imaging techniques discussed (Fluorescence imaging, Super-resolution imaging, Photoacoustic imaging, and Raman imaging) and discussing the key points of each technique, including their pros and cons or other details.

10.   In Section 2.3.1, the authors have briefly mentioned some reports on CPs without providing any specific details such as polymer names or relevant information. This may cause confusion for readers as they won't be able to understand the context fully. For instance, the sentence "Wang et al. designed a CPs treatment system for targeted delivery of drugs to mitochondria for collaborative anti-cancer treatment" lacks crucial information such as the name of the polymer, the type of cancer targeted, and the mode of action. This issue is present in other parts of the article as well. Please consider adding some concluding remarks to the end of the paragraph. This would help provide a clear summary of the main points discussed and give readers a sense of closure.

11.   Could you please clarify whether the term "Collaborative therapy" mentioned in section 2.3.5 is commonly used in the field of cancer therapy? A Google search did not yield any results related to cancer therapy, but instead showed information about a different technique that is commonly used in the field of psychology. Therefore, it may be beneficial to either avoid using this term or provide some references where it has been used in the context of cancer therapy.

·         Certain sentences, such as those found in P3, L108-109, are difficult to understand. Please revise them for clarity.

·         Capital letters should not be used in the middle of sentences, such as in "Fluorescence" in P3 L125. Please review the entire manuscript for these issues.

·         Please review and proofread the manuscript for English language corrections.

Round 2

Reviewer 1 Report

It is deserved for publication without needs to more revisions.

Reviewer 2 Report

please address the below comments for the manuscript entitled "Conjugated polymeric materials in biological imaging and cancer therapy": 

1- All figures, specifically figure 6 have not suitable quality. Authors should improve them before more considering this manuscript.

2- At the end of the introduction, please explain innovative aspects of your study compared to previous studies.  

3- The conclusion section can be reorganized based on the main findings of the study.
4- The manuscript would benefit from English grammar editing.

The manuscript would benefit from English grammar editing.
